# Carbon nanotube-based three-dimensional monolithic optoelectronic integrated system

Yang Liu[1], Sheng Wang[2], Huaping Liu[3,4] & Lian-Mao Peng[1,2]

Single material-based monolithic optoelectronic integration with complementary metal oxide semiconductor-compatible signal processing circuits is one of the most pursued approaches in the post-Moore era to realize rapid data communication and functional diversification in a limited three-dimensional space. Here, we report an electrically driven carbon nanotube-based on-chip three-dimensional optoelectronic integrated circuit. We demonstrate that photovoltaic receivers, electrically driven transmitters and on-chip electronic circuits can all be fabricated using carbon nanotubes via a complementary metal oxide semiconductor-compatible low-temperature process, providing a seamless integration platform for realizing monolithic three-dimensional optoelectronic integrated circuits with diversified functionality such as the heterogeneous AND gates. These circuits can be vertically scaled down to sub-30 nm and operates in photovoltaic mode at room temperature. Parallel optical communication between functional layers, for example, bottom-layer digital circuits and top-layer memory, has been demonstrated by mapping data using a $2 \times 2$ transmitter/receiver array, which could be extended as the next generation energy-efficient signal processing paradigm.

[1] Key Laboratory for the Physics and Chemistry of Nanodevices and Academy for Advanced Interdisciplinary Studies, Peking University, Beijing 100871, China. [2] Key Laboratory for the Physics and Chemistry of Nanodevices and Department of Electronics, Peking University, Beijing 100871, China. [3] Beijing National Laboratory for Condensed Matter Physics, Institute of Physics, Chinese Academy of Sciences, Beijing 100190, China. [4] Collaborative Innovation Center of Quantum Matter, Beijing 100190, China. Correspondence and requests for materials should be addressed to L.-M.P. (email: lmpeng@pku.edu.cn).

With the release of the 2015 International Technology Roadmap for Semiconductors (ITRS), the semiconductor industry formally acknowledged the fact that Moore's law is nearing its end[1]. Although many proposals embrace completely different paradigms, none have been demonstrated at a system level except for the carbon nanotube (CNT) computer[1,2]. Nevertheless, two approaches are being actively pursued, namely, using on-chip optical communications and a three-dimensional (3D) architecture[1–9]. Compared with pure electrical or optical circuits, versatile-functionality optoelectronic integrated circuits (OEICs) can capitalize on not only the high bandwidth and parallelism of optical transmission but also the electrical input–output isolation. Numerous studies have been conducted over the past few decades using III–V, II–VI, Ge, semiconductor nanowires and a burgeoning class of two-dimensional (2D) materials to realize optoelectronic active or passive functional modules[10–16]. However, various materials are required for monolithic OEICs, and combining these materials and modules of different functionalities in a 3D geometry with a high integration density has proved challenging because of the fabrication conflicts between the electronics and photonics based on conventional semiconductors[10]. Furthermore, in fibre-optic communications, the highest bandwidth is limited because of the finite different wavelength patterns, for example, 16 channels for single-fibre coarse wavelength division multiplexing[10,17]. Therefore, it is significant to realize a single material-based monolithic OEIC via a complementary metal oxide semiconductor (CMOS)-compatible low-temperature fabrication process to enable 3D functionality of unlimited bandwidth at low cost[6,10,18,19]. Semiconducting single-walled CNT (s-SWCNT) is a direct-bandgap material with an ultrathin body of 1–3 nm, an extremely high carrier mobility, and a broadband response from 0.2 to 1.5 eV; these features make s-SWCNTs ideally suited for use in electronic and optoelectronic devices and multilayer stacking in a 3D geometry[20–24]. Thermal-type CNT vertical integration has been studied in off-chip fashion and CNT/Si heterojunction digital optoelectronics has been discussed as well, indicating that there is plenty of room for CNT optical interconnection[25,26].

Here, we demonstrate that energy-efficient nanoelectronic and optoelectronic subsystems, such as receivers and transmitters, for creating on-chip electrically driven miniaturized OEICs, can be fabricated on the same footing using CNTs via a CMOS-compatible low-temperature doping-free technique, and optical communication between stacked functional layers, such as microprocessor and memory layers, can be realized via parallel channel-division multiplexing (CDM) mapping with CNT-based 3D OEICs.

## Results

**CNT receiver and transmitter**. *CNT photovoltaic receiver*. A receiver is typically composed of a photodetector and signal processing circuit, which are usually made of different materials, for example, Ge for the photodetector and Si for the signal processing circuit[10]. Thus, different fabrication processes of distinct materials make monolithic integration a challenging task. However, nanoelectronic and optoelectronic devices can be readily integrated together using s-SWCNTs by simply selecting symmetric contacts for CMOS devices and asymmetric contacts for optoelectronic devices[21–24]. Here, we combine contact engineering with high-semiconducting-purity CNT films to construct photovoltaic receiver[23] and amplify the photovoltage[24] *in situ* to significantly improve its responsivity. Numerous investigations have also been undertaken to develop

CNT thermal detectors[27], but typical device size is large and system requirement is high, preventing the use of thermal detectors in OEICs.

A CNT-based receiver (Fig. 1a) consists of a CNT cascading detector and an n-type field-effect transistor (FET) for an analogue switch or, more generally, signal processing (the specific characterizations of the s-SWCNTs used are given in Supplementary Fig. 1). A false-colour scanning electron microscopy (SEM) image showing a real receiver is given in Fig. 1b. However, unlike in Fig. 1a, where for clarity only a two-cell cascading detector is shown, in the real receiver, a nine-cell cascading detector (channel length: $L = 2\,\mu m$, channel width: $W = 30\,\mu m$) is used to increase the signal and the signal-to-noise ratio, and an interdigital device structure is used for the n-FET ($L = 4\,\mu m$, $W = 30\,\mu m$) to provide a sufficiently large driving capability for additional CMOS processing circuits. When illuminated with infrared (IR, with $\lambda = 1,800\,nm$) light, the detector generates a photovoltage (Fig. 1c), which logarithmically depends on the light intensity (Supplementary Fig. 2). The cascading detector indeed acts as a first-stage linear amplifier, and the output of the amplifier can be tuned by changing the number of cascading modules in the channel (Fig. 1d) from 0.26 V for a single-cell detector to 2.3 V for a nine-cell detector. The CNT-based n-FET (inset of Fig. 1e) can be fabricated with Sc contacts, giving rise to a large on-state current of 1.06 mA at $V_{ds} = 3\,V$ (Fig. 1f) and a high on/off ratio of over $10^5$ (Fig. 1e and Supplementary Table 1). When the cascading detector is connected with the n-FET (as shown in the inset of Fig. 1h), the detector functions as an optical gate for the n-FET. Controllable photovoltage can be generated on the gate of the FET by tuning the incident light intensity (Fig. 1g), with an on-state current of 1.13 mA which is comparable to that of the electrically controlled FET. The most important performance figure of merit for a receiver is the responsivity, which can be tuned monotonously by the bias on the drain with a maximum responsivity of $0.67\,A\,W^{-1}$ (Fig. 1h, Supplementary Table 1). Because of the low dark current and incident power density-dependent current output, the specific electrical and optical on/off ratio can both reach $\sim 10^5$ (Supplementary Table 1). The maximum photovoltage responsivity and detectivity of the cascading detector can also be evaluated to be $\sim 10^8\,V\,W^{-1}$ and $10^{11}$ Jones respectively (detailed calculations are given in Supplementary Note 2), and the detector responses in the whole near IR band, that is, from 1,165 to 2,100 nm (ref. 23). The dynamic response of the receiver (Fig. 1i) is obtained via chopped incident light illumination. This receiver can be utilized as either an independent high performance subsystem or as an important component in an OEIC.

*CNT transmitter*. Photons are bosons that travel at the speed of light, making them difficult to manipulate[10]. Therefore, an on-chip electrically driven light source is quite important for monolithic OEICs, where photons in the communication band of 1,310 or 1,550 nm can be easily manipulated by controlling electrons to form a transmitter, which is an essential part in an OEIC system[10,28]. Furthermore, an on-chip light source may lead to a higher integration density and lower power consumption compared to that of an off-chip light source, in which energy is wasted in terms of coupling losses and proportional usage[28]. A CNT transmitter (Fig. 2a) can be constructed by connecting a controlling n-FET ($L = 4\,\mu m$, $W = 30\,\mu m$) with an emitter ($L = 0.5\,\mu m$, $W = 30\,\mu m$); and a real transmitter is shown by the false-colour SEM image in Fig. 2b. The controlling n-FET functions as a digital modulator to tune the emission intensity of the CNT emitter, and a typical electroluminescence (EL) spectrum from the emitter is shown in Fig. 2c and can be fitted with two Lorentzian functions and attributed to the emissions of (8, 3) and (8, 4) CNTs

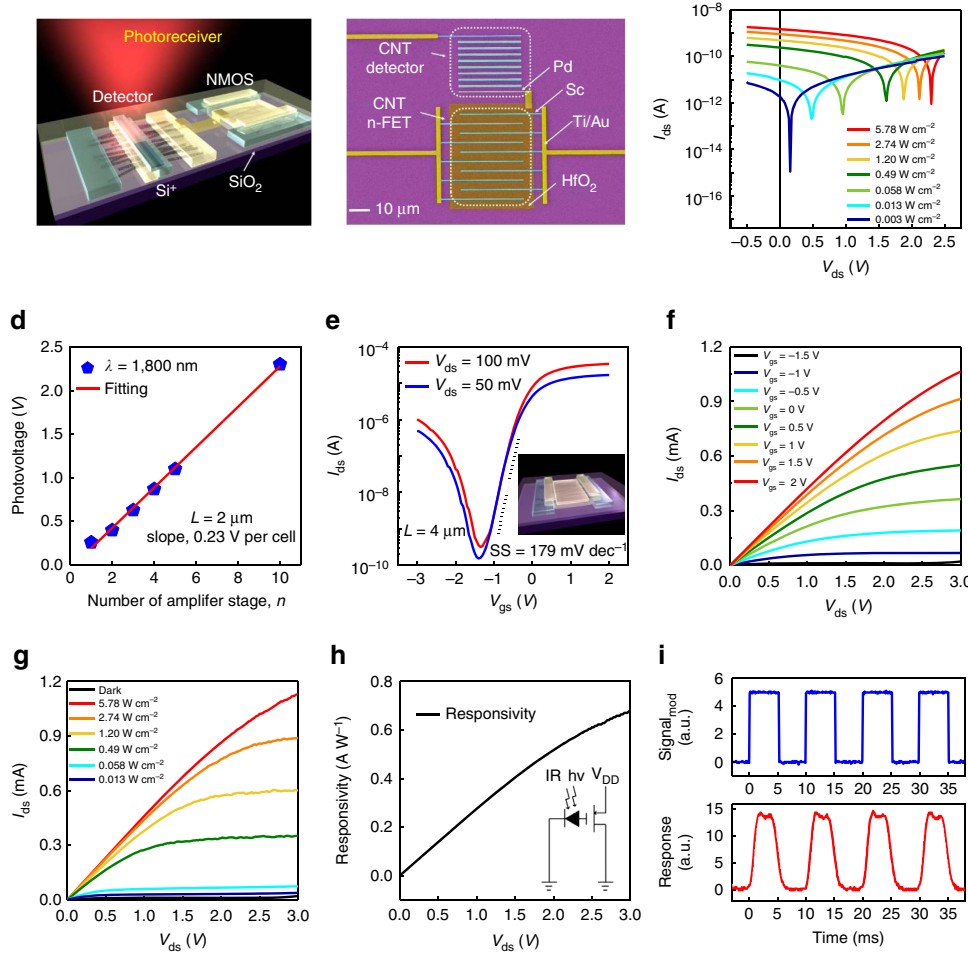

**Figure 1 | Structure and performance of the photovoltaic CNT receiver.** (**a**) Schematic of the photovoltaic CNT receiver, consisting of a first-stage cascading linear amplifier and a second-stage analogue switch. (**b**) False-colour SEM image of the real photoreceiver, consisting of a nine-cell cascading photodetector and interdigital n-FET (field effect transistor; scale bar, 10 μm). (**c**) Photoresponse of the cascading first-stage amplifier. (**d**) Linearity of the first-stage amplifier versus the amplifier stage number. (**e**) Set of transfer characteristics of the CNT film n-FET with different bias voltages. (**f**) Output characteristics of the n-FET in **e** measured with $V_{gs}$ varying from 2 to −1.5 V. Inset is the schematic top-gate n-FET structure. (**g**) Output characteristics of the photovoltaic receiver in the dark and under various incident IR illuminations from 5.78 to 0.013 W cm$^{-2}$. (**h**) Photocurrent responsivity of the receiver. Inset is the corresponding equivalent circuit diagram. (**i**) Dynamic time trace of the receiver (power density: 5.78 W cm$^{-2}$, wavelength: 1,800 nm, light source: NKT super-continuous spectrum laser).

(detailed characterizations are shown in Supplementary Fig. 3) at 0.92 eV and 1.04 eV, respectively, with a full width of half-maximum of 92 and 103 meV, respectively (see also Supplementary Fig. 4). The integrated emission intensity increases exponentially with applied bias, indicating an impact excitation mechanism-dominated illumination process (Fig. 2d). When compared with the corresponding photoluminescence spectrum (Fig. 2e), EL peaks have been red-shifted by ∼171 meV and 242 meV for the (8, 3) and (8, 4) peaks, respectively, which can be attributed to the trion emission mechanism[29,30]. The trion mechanism red-shifts the emission towards the telecommunication band of 1,330 nm and lightens up the otherwise dark excitons[10,20], making the overall emission sufficiently high to be detected by a nearby CNT detector (as discussed below). The use of chirality-selected CNT films in the emitter narrowed the generally broad EL emission from non-chirality-selected CNT film-based emitter (Supplementary Fig. 6a). In general, the emission wavelength can be readily controlled via the selection of appropriate CNTs with suitable chirality, which can tune the communication wavelength of the transmitter.

The CNT transmitter (Fig. 2a) may be simplified as the equivalent circuit shown in the inset of Fig. 2f. In general, the intensity of a CNT emitter is proportional to the driving current, which can be tuned over three orders of magnitude by changing the gate voltage on the controlling n-FET (Fig. 2f). The n-FET can be readily switched between the on-state (with $V_{gs}$ = 5 V) and off-state (with $V_{gs}$ = − 2 V), corresponding to the high and null EL emissions respectively (Fig. 2g). When a modulated gate voltage is applied, the transmitter can be switched between the on (or emission) state and off (or null emission) state (Fig. 2h), which can be used to construct on–off keying intensity modulation via electronic circuits to control the n-FET. Experimentally, there exists a threshold electrical field of 1.2 V for the transmitter to emit due to the impact excitation mechanism. Thus, the emission state of the transmitter can also be readily controlled by varying the bias to switch between null emission ($V_{ds} < 1.2$ V) and emission ($V_{ds} > 1.2$ V).

**Heterogeneous optoelectronic circuits.** In the post-Moore era, chip development is expected to be more closely influenced by the

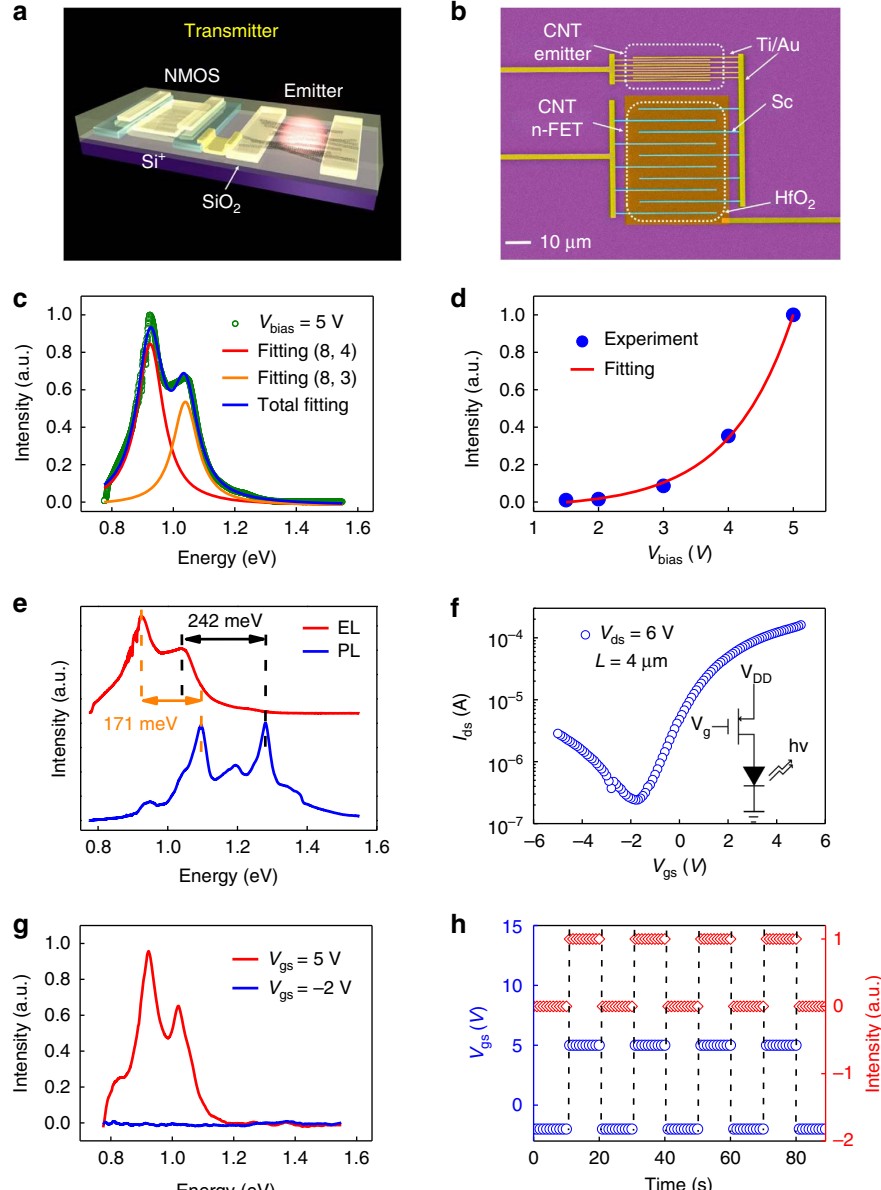

**Figure 2 | Structure and performance of the CNT transmitter.** (**a**) Schematic of the CNT transmitter consisting of a digital CNT n-FET (field effect transistor) and emitter. (**b**) False-colour SEM image of the real CNT transmitter (scale bar, 10 μm), consisting of an interdigital n-FET and emitter. (**c**) EL spectrum and Lorentzian fitting. (**d**) Integrated IR emission intensity as a function of voltage bias. (**e**) Corresponding EL and photoluminescence spectra. (**f**) Transfer characteristics of the CNT transmitter with a modest $V_{ds} = 6$ V. Inset is the corresponding equivalent circuit diagram. (**g**) Tunable and controllable IR emission of the transmitter. The two EL emission spectra correspond to $V_{gs} = -2$ and 5 V, respectively. (**h**) Time trace of the transmitter.

specific demands of consumers; therefore, versatile chips with multi-functionality are becoming increasingly important[1,7]. As part of these efforts, heterogeneous optoelectronic logic gates have attracted worldwide attention due to their potential for combining optical and electrical advantages to provide such functions as optical parallel transmission and electronic input–output isolation. It is significant for these systems that controllable manipulation of the applied bias and IR light coexist either in the input or output to form heterogeneous logic[31]. A receiver can be configured to function as an AND gate (Fig. 3a) under the following conditions: the IR incident light is input 1, the bias on the drain of the controlling n-FET is input 2 and the current flow $I_{out}$ in the source is the output. According to the truth table of the AND logic gate (inset of Fig. 3a), the output is in logic 0 when input 1 is 0 W cm$^{-2}$ and input 2 is 0 V, and the output is in logic 1 when input 1 is 5.78 W cm$^{-2}$ and input 2 is greater than 0 V (for example, 3 V).

One such an experimental realization is illustrated in Fig. 3b, showing that when input 1 and input 2 are both in logic 1, the output generates a current of $I_{out} \cong 1$ mA or logic 1. In contrast, when either input 1 or input 2 remains in the logic 0 state, the output generates no detectable current $I_{out}$ or logic 0. Typical waveforms of $I_{out}$ are shown in Fig. 3c, verifying the heterogeneous AND logic gate behaviour. Similarly, a heterogeneous AND gate can also be realized by configuring a CNT transmitter (Fig. 3d). Two electrical inputs on the source and gate ends of the controlling n-FET are defined as input 1 and input 2, respectively, and the IR emission from the CNT emitter is defined as the output. The truth table for the transmitter AND gate (inset of Fig. 3d) specifies logic 0 of the inputs as the off-state of the n-FET with $V_{ds} = 0$ V and $V_{gs} = -2$ V for input 1 and input 2, respectively; logic 1 of the inputs are specified to be $V_{ds} > 1.2$ V (for example, 3 V) and $V_{gs} = 5$ V for input 1 and input 2, respectively. The current output

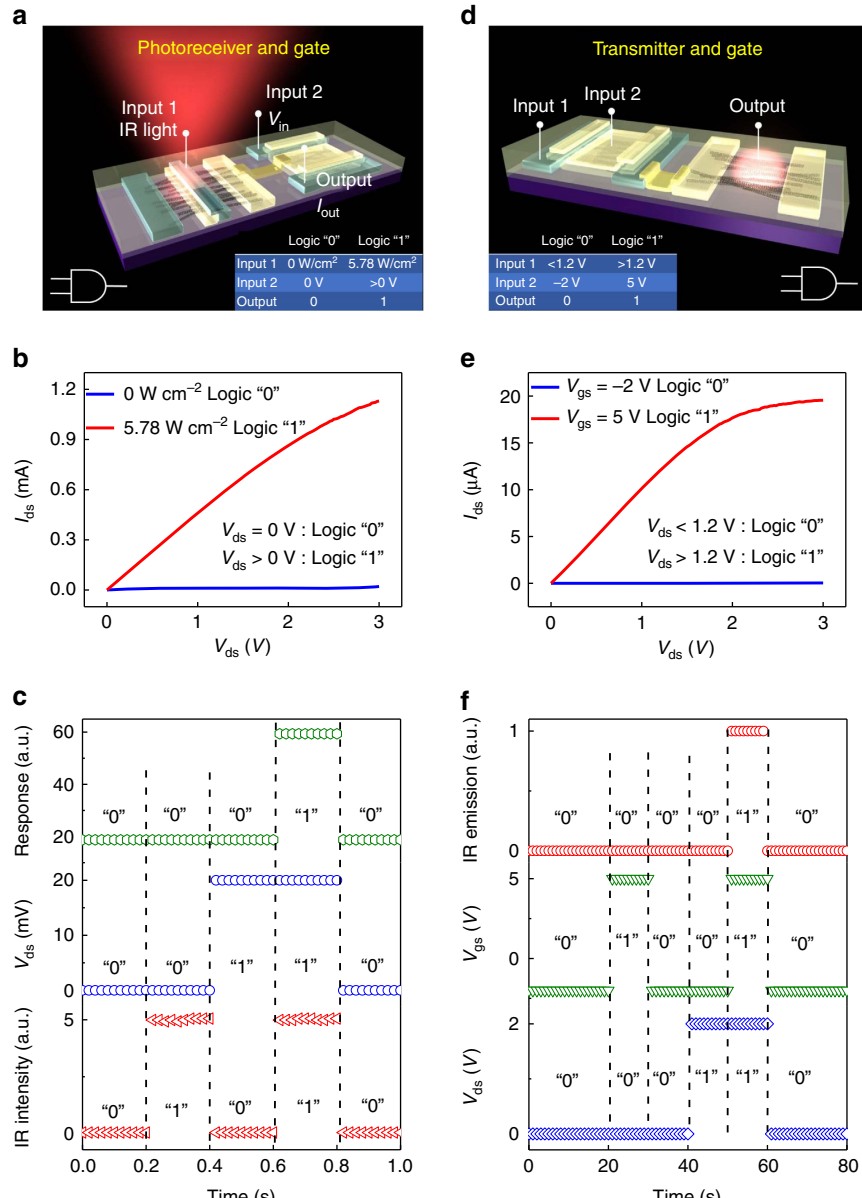

**Figure 3 | Heterogeneous optoelectronic logic gate.** (**a**) The AND gate realized via a photovoltaic receiver with optical or electrical inputs and output. The left inset is the equivalent circuit. The right inset is the truth table defining the logic 0 and 1 states for both the input and output, as determined by the output characteristics shown in **b**. (**c**) Waveform of the AND gate. (**d**) The AND gate realized via a transmitter with electrical inputs and optical output. The left inset is the truth table corresponding to the output characteristics shown in **e**. The right inset is the equivalent circuit. (**f**) Waveform of the AND gate.

of the n-FET (Fig. 3e) can be driven to a sufficiently high value when both input 1 and input 2 are in logic 1, leading to large IR emission or logic 1 for output; if either input 1 or input 2 is in logic 0, then the output IR emission is null or in logic 0. The nearly perfect output waveforms (Fig. 3f) verify clearly the truth table of the AND gate. In this manner, both optical on-chip emission and electrical input–output isolation can be integrated in an OEIC subsystem, providing the potential to construct 3D OEICs.

**3D optoelectronic integrated circuit**. Monolithic 3D integration, that is, stacking various functional modules on top of each other, has been widely pursued to further increase the integration density and create versatile functionality per chip area in the post-Moore era[1,3,6,7,19]. Here, we consider the stacking of the CNT transmitter and receiver, aiming to realize the parallel

transmission of data between functional layers. As shown in Fig. 4a, two CNT emitter channels and a two-cell cascading detector are vertically integrated with a 20-nm HfO₂ layer in between to provide electrical isolation (corresponding side-view was shown in the inset of Fig. 4e). The real system used in this work is somewhat more complex, with a nine-channel interdigital emitter in the top layer and a well aligned nine-cell cascading detector in the bottom layer (Fig. 4b, corresponding alignment process is discussed in Supplementary Note 6 and shown in Supplementary Fig. 5). The top-layer emitter is based on chirality-selected (8, 3) and (8, 4) CNTs, and the bottom-layer detector is fabricated using high-semiconducting-purity CNTs. Typical EL spectra obtained from the top-layer emitter at various biases (Fig. 4c) indicate that the emission of the top-layer emitter falls between the $E_{11}$ and $E_{22}$ absorption peaks of the bottom-layer detector (Supplementary Fig. 6b), making the

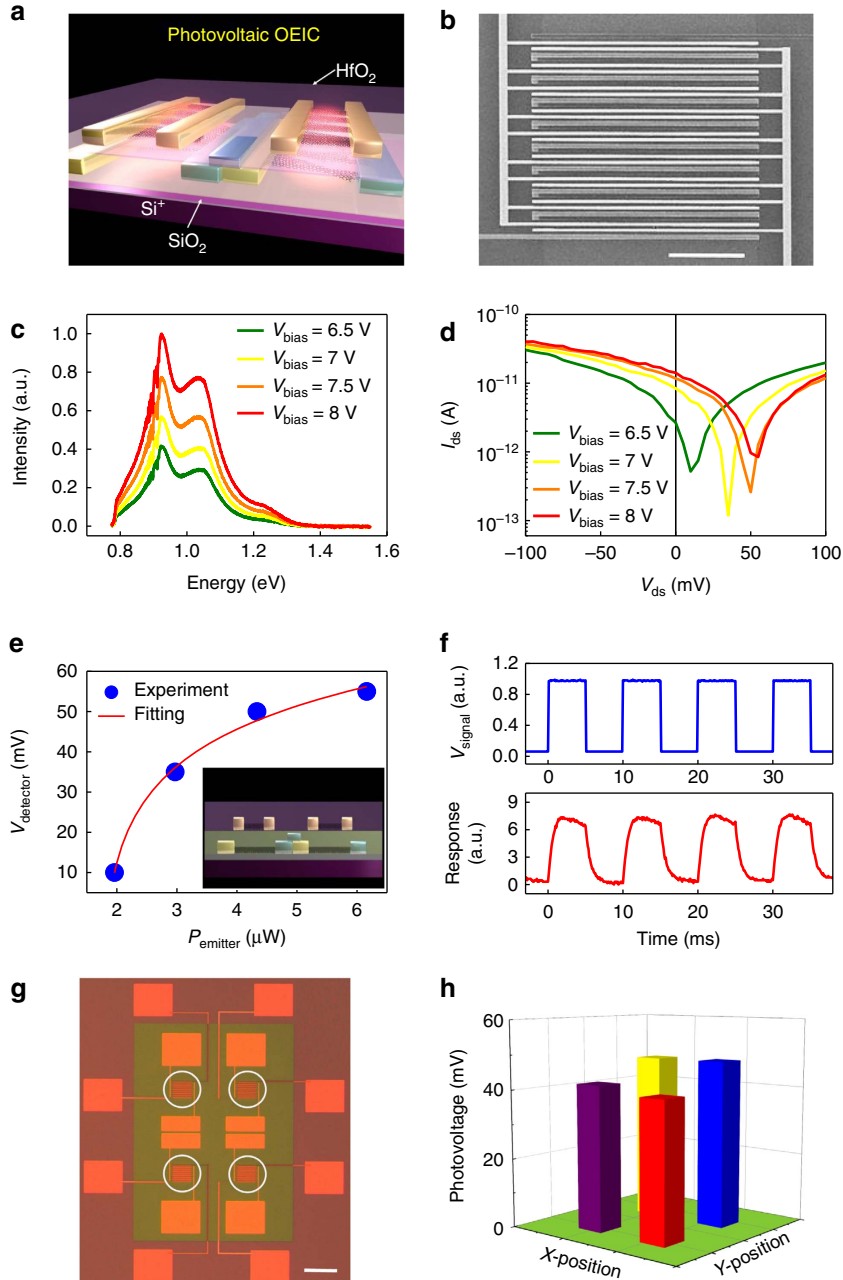

**Figure 4 | On-chip vertical near-field OEIC.** (**a**) Schematic of the vertical near-field OEIC, consisting of a top-layer emitter and a bottom-layer cascading detector. (**b**) SEM image of the vertical OEIC (scale bar, 10 μm). (**c**) EL spectra of the top-layer emitter. (**d**) Corresponding output characteristics of the bottom-layer photovoltaic detector. (**e**) Photovoltage of the bottom-layer detector versus the illumination power of the top-layer CNT emitter. Inset is the front-view graph of **a**. (**f**) Dynamic time trace of the 3D OEIC. (**g**) Optical image of the 2 × 2 vertical parallel-transmission array (scale bar, 50 μm). Each white circle denotes an active CDM channel region. (**h**) Corresponding parallel-transmission mapping result of the 2 × 2 CDM mapping array.

emission from the top layer to be detectable by the bottom-layer detector. The chirality-selected CNT films for the top-layer emitter are *in situ* fabricated on each channel of the bottom-layer cascading detector via the ac dielectrophoresis technique[32], which is a low-temperature process and compatible with our doping-free technique. Thus, CNT-based 3D OEICs can be readily fabricated in a CMOS-compatible low-temperature fabrication line (more details are given in Supplementary Note 3 and Supplementary Table 2)[19].

The entire height of the active channel of our 3D OEIC is sub-30 nm, which is in the near-field region of the top-layer CNT emitter[26,33–35]. The emitted photons from the top-layer emitter

are partially absorbed by the bottom-layer cascading detector, and the other part propagates upward into the free space and can thus be detected with an off-chip InGaAs detector simultaneously with the electrical measurements on the bottom-layer receiver. The absorption of the light emission in the bottom-layer detector results in a photovoltaic response. When compared with well separated emitter and detector pair, the performance of the system remains basically the same after stacking them together (detailed discussions are given in Supplementary Note 7 and Supplementary Fig. 7). The output characteristics of the CNT detector (Fig. 4d) show that photocurrent and photovoltage both increase with increasing voltage applied to the top-layer emitter;

in addition, the measurement is simultaneously calibrated with upward emitted photons using an external InGaAs detector (more details are given in Supplementary Note 4 and Supplementary Fig. 8). The radiated energy flow ratio between the bottom and top edges (more details are given in Supplementary Note 8 and Supplementary Fig. 9) is calculated to be ∼2 using the finite element method, and thus, the bottom edge energy can be estimated using the experimentally recorded EL spectra from the top of the emitter. The photovoltage of the bottom-layer CNT detector is logarithmically related to the emission power from the top-layer emitter (Fig. 4e), suggesting that the detector is operating in photovoltaic mode, which is faster and more efficient than thermal detectors based on bolometric effect in principle[36]. Furthermore, in situ Raman spectra were recorded with an off-chip silicon detector working synchronously with electrical voltage stimulation (Supplementary Fig. 10a). Although the Raman G-peak position is generally sensitive to the CNT temperature and thus to the bias on the emitter, the Raman G-peak (Supplementary Fig. 10b) remains unchanged during electrical measurement, confirming that the emitter operates as a photon emitter instead of as a blackbody emitter[28,35,36]. Indeed this cold emitter is more desirable in an OEIC compared with a hot blackbody thermal emitter with a heating temperature of up to hundreds of degrees centigrade, which will increase the power consumption of the system, thereby limiting its integration density[28,36]. The dynamic response of the system is also measured and given in Fig. 4f, showing that the photovoltage response of the bottom-layer detector correctly follows the square-wave voltage bias applied on the top emitter. However, the system response is limited by parasitic capacitances and mismatch with external measurement systems (more analyses are given in Supplementary Note 5). The emission speed of a CNT emitter can be extremely high (∼140 ps), which allows for a communication rate of up to ∼10 Gbps (ref. 36), and the demonstrated response speed of a CNT detector is even higher, suggesting that high-speed communication of up to tens of Gbps should be possible for CNT-based OEICs with an optimized layout[36–38]. Furthermore, the transfer characteristics of the n-FET of the bottom-layer receiver were also measured under similar illumination conditions as that of the detector, that is, between 1.19 and 5.92 μW, showing no difference with that measured in the dark (more analyses are given in Supplementary Note 9 and Supplementary Fig. 11a). This result suggests that the emitted photons from the top-layer emitter do not significantly heat the conduction channels of active devices in the bottom layer, verifying that the detected signal is entirely due to the photovoltaic effect rather than bolometric or photothermoelectric effects and that the performance of the electronic devices and circuits on the bottom functional layer is not affected (more analyses are given in Supplementary Note 9 and Supplementary Fig. 11b). Thus, optical and electrical isolation is automatically realized in our system, and stray light in our 3D OEIC does not affect the performance of the CMOS devices involved.

## Discussion

It is anticipated that next-generation information technology will process unprecedented amounts of data, and 3D integration may improve the energy efficiency of abundant data applications[3]. In principle, CNT OEICs may be used for inter-layer data communication by modulating light intensity via, for example, on–off keying, as illustrated in Supplementary Fig. 13a. Signal or data stored in the top layer can be mapped using a 2D array of transmitters/receivers directly to the bottom layer via CDM. Unlike traditional semiconductor-based OEICs, all CNT electronic and optoelectronic devices can be monolithically integrated in 3D configuration (Supplementary Fig. 13b). As succinctly shown in Supplementary Fig. 13b, the top functional layer consists of memory cells and a 2D array of transmitters, and the bottom functional layer consists of a 2D array of receivers and digital circuits (for example, CPU). The optical image in Fig. 4g shows a real OEIC system with a $2 \times 2$ array of transmitter/receiver pairs in the middle (marked with white circles) for vertical data mapping. In this OEIC system, each transmitter/receiver pair consists of 10 channels to increase the signal-to-noise ratio, and each channel occupies $12 \, \mu m^2$. In principle, such a channel size may be further reduced when better CNT films become available, for example, parallel arrays of CNTs with a higher density and purity. Currently, the typical pitch size of a CNT photodiode has been scaled down to sub-50 nm (ref. 39). Top-layer data (0 and 1 binary signal strings) stored in the memory cells can be shifted to the 2D array of transmitters, mapped to the 2D array of receivers in the bottom layer in parallel and then shifted to adjacent digital circuits (for example, CPU) for logic operations (Supplementary Fig. 13b). The demonstrated data mapping of a $2 \times 2$ CDM array (Fig. 4h) generates relatively uniform photovoltage of 48, 42, 40 and 48 mV for the 4 bottom-layer receivers. In principle, the upward emission can be suppressed using photon isolation layer such as 100-nm Ti to avoid interference between emitters during the mapping process (further analyses are discussed in Supplementary Note 10 and Supplementary Fig. 12). As discussed earlier, the single-channel transmission speed of the transmitter/receiver pair could reach 10 Gbps, and the communication array size may be readily scaled up to any desired size with $n \times n$ separated channels, enabling a communication speed of $10n^2$ Gbps due to the parallel data transmission characteristics of CDM. This scheme makes data transmission between functional layers more efficient than traditional fibre-optic communications, whose highest bandwidth is restricted because of the finite different wavelength patterns, for example, only 16 channels for a single-fibre coarse wavelength division multiplexing[10–17]. It potentially enables the realization of high-speed and large-scale monolithic multilayer OEICs.

To summarize, we have realized a monolithic 3D OEIC system, consisting of arrays of photovoltaic receivers, electrically driven transmitters and CMOS signal processing circuits, all of which are fabricated using CNTs via a CMOS-compatible doping-free technique. Diversified functions can also be realized via heterogeneous gates, such as the AND gate. This work provides a different paradigm for parallel data communication between stacked functional layers, for example, microprocessor layer and memory layer, which could lead to an inter-layer data communication speed of up to $10n^2$ Gbps when an $n \times n$ transmitters/receivers array is used.

## Methods

**CNT film fabrication.** CNT films were deposited via a liquid-phase method[23]. The chirality-selected (8, 3) and (8, 4) CNTs are purified via agarose gel chromatography, and such CNTs are assembled in situ via a dielectrophoresis technique[32]. The morphology of the CNT films was examined using SEM and atomic force microscopy.

**Device fabrication.** For all devices, the device layout patterns were defined using electron-beam lithography (EBL, Raith150-two, Germany) and all metal films were deposited using electron-beam evaporation (EBE, Kurt J. Lesker) with a standard lift-off process. For n-FET and cascading detector, CNTs outside the active channels were etched using reactive ion etching (RIE, Trion Minilock-Phantom III). The insulating layer was deposited using low-temperature atomic layer deposition (Cambridge, ALD). Detailed fabrication processes are listed in Supplementary Note 1.

**Device characterization.** In this work, 488-nm continuous wavelength laser was used for Raman characterization (Jobin Yvon/Horiba Company). A super-continuous spectrum laser (NKT Company) was used to characterize the response of the receiver, and the IR wavelength of the laser can be tuned from 1,165 to 2,100 nm. Optoelectronic measurements on CNT devices were performed on a probe station attached to the Horiba system. All electronic transportation measurements were performed using a Keithley 4,200 semiconductor analyzer at room temperature under ambient conditions. A square-wave electronic signal was generated via a waveform generator (Agilent 33220A), and the output signal of the voltage amplifier (SR560) was detected and recorded using an oscilloscope (Agilent DSO7054A). The input optical signal was modulated using a SR540 chopper.

**Data availability.** The data that support the findings of this study are available from the corresponding author upon request.

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

## Acknowledgements

This work was financially supported by the Ministry of Science and Technology of China (Grant No 2016YF0201902), National Natural Science Foundation of China (No's 61621061, 61427901, 61390504 and 51472264) and the Beijing Municipal Science and Technology Commission (Grant No D161100002616001-3).

## Author contributions

L.-M.P. proposed and supervised the project, Y.L. fabricated devices and performed all electric and optoelectronic experiments, and H.L. performed separation and processing of (8, 3) and (8, 4)-enriched CNTs. L.-M.P. and Y.L. analysed the data and co-wrote the paper, and all authors discussed the results and commented on the manuscript.

## Additional information

**Competing interests:** The authors declare no competing financial interests.

**Publisher's note**: 

