## [Peer Review File · Nature Communications]

Reviewers' comments:

Reviewer #1 (Remarks to the Author):

In this paper, the authors present the fabrication of an electrically-driven carbon nanotube (CNT)-based on-chip 3D optoelectronic integrated circuit (OEIC), including photovoltaic receivers, electrically driven transmitters and on-chip CMOS circuits. Parallel optical communication between functional layers, e.g. bottom-layer CMOS-based digital circuits and top-layer memory, is demonstrated by mapping data using a 2×2 transmitter/receiver array via channel division multiplexing (CDM). The design and results are interesting and may be useful for fabricating next-generation 3D OEIC system, including the enabling of new application spaces that were previously inaccessible. Upon addressing the following important points, mostly for clarification, I do recommend the manuscript for publication:

- 1) The structure, fabrication details and dimensions for all the devices should be supplied in the paper, such as channel length, width, etc. In addition, the schematics, while very flashy, are actually very easy to follow and could use more scientific content (perhaps one flashy one and the rest more scientifically detailed?).
- 2) On line 79, "for clarity" is repeated twice, should remove one from the text.
- 3) As for the fabrication of the vertically stacked devices, how were the patterns/layers aligned? Is there any impact from overlapping and non-overlapping regions? A comparison should be provided between the performance of the single layer devices and how they change (if at all) once they are stacked together. From Figure 4d, it seems the performance becomes worse. Please explain.
- 4) More details on the FEM simulation on page 9 need to be provided, perhaps in the supplementary information.
- 5) On page 10, the authors stated that "the system response is limited by parasitic capacitances and mismatch with external measurement systems...", please explain the details on this.
- 6) It is described that "the emitted photons from the top layer emitter do not heat the conduction channels ... not affected." Does this remain true if running for a longer time? Any further data to support this conclusion?
- 7) In the double-layer 3D OEIC system shown in Fig.5, how to avoid the interference between emitters during the mapping process? Should the insulating layer be transparent for this design? In addition, it would be interesting to know if this kind of design works for more than two layers?
- 8) Equivalent circuits for the AND in Fig.3 should be added into the figures.

Reviewer #2 (Remarks to the Author):

Y. Liu et al. reports a 3D optoelectronic integrated circuit fabricated with carbon nanotube dopant free electronic and optoelectronic devices. They demonstrate CNT PV receiver, a CNT IR transmitter, heterogeneous integration with logic CNT elements, and the monolithic 3D system integration.

I believe that the overall effort merits publication in Nature Communications. I am concerned about the claim in the communication speed of 10n2 Gb/s. I suggest this to be removed from the

abstract. It is clear that the system parasitics dominate the performance of the device at kHz frequencies that are tested in this work. For 3D integrated OEIC, the engineering challenges reside in interconnect bandwidth and power. These two challenges were not addressed in this work and as such, while the fabrication and integration is exotic, I am afraid that the work would be another demonstration that 'we can do it also with CNTs' without a strong and complete benchmarking of performance metrics against existing technologies.

I recommend publication with minor revision.

Reviewer #3 (Remarks to the Author):

The authors have prepared an electrically driven carbon nanotube (CNT)-based on-chip 3D optoelectronic integrated circuit (OEIC), which show diversified functionality such as the heterogeneous "AND" gates. This work seems to be interesting and the structure is well organized, but CNT foam structure has well-studied before, and there is not enough novelty and the results do not show significant findings.

there are also some questions needed to be further clarified.

1. What is the advantage of single material (CNTs)-based monolithic optoelectronic compared with CNTs-Silicon based monolithic optoelectronic?
2. Please list the photocurrent responsivity, electrical on/off ratio, and optical on/off of the device in this work. In order to clearly reveal the excellent performance of the as-prepared device, please compare those parameters with the past reports.
3. The structural and scalability are very important for the practical application of the device. How is the method presented in this work?
4. The authors just show a 2x2 array of transmitter/receiver pairs, but what is the 4x4 array of transmitter/receiver pairs? If possible, please provide the corresponding performances.
5. The infrared wavelength of the laser is just from 1165 to 2100 nm, so what is the sensitivity/performance of this device in variable light conditions (the laser with wide wavelength)?
6. The authors claimed that the emission wavelength can be readily controlled via the selection of appropriate CNTs with suitable chirality. Maybe it means that it will have different performance/conclusion if different CNTs have been used.

Point-to-point response to the referees' comments:

Reviewer# 1

In this paper, the authors present the fabrication of an electrically-driven carbon nanotube (CNT)-based on-chip 3D optoelectronic integrated circuit (OEIC), including photovoltaic receivers, electrically driven transmitters and on-chip CMOS circuits. Parallel optical communication between functional layers, e.g. bottom-layer CMOS-based digital circuits and top-layer memory, is demonstrated by mapping data using a 2×2 transmitter/receiver array via channel division multiplexing (CDM). The design and results are interesting and may be useful for fabricating next-generation 3D OEIC system, including the enabling of new application spaces that were previously inaccessible. Upon addressing the following important points, mostly for clarification, I do recommend the manuscript for publication:

1. The structure, fabrication details and dimensions for all the devices should be supplied in the paper, such as channel length, width, etc. In addition, the schematics, while very flashy, are actually very easily to follow and could use more scientific content (perhaps one flashy one and the rest more scientifically detailed?).

Response: Accepted with thanks.

➤ **Device structure and dimensions:**

Devices used in this work can be classified into three types, i.e. CNT cascading detector, CNT emitter, and CNT n-FET as shown in Fig. R1 (below). The channel length and width can be defined as L and W, respectively. A CNT cascading detector consists of nine cells (L=2 μm, W=30 μm, see Fig. R1a), and the interdigital CNT emitter consists of ten fingers (L=0.5 μm, W=30 μm, see Fig. R1b). The CNT n-FET also used ten-finger structure to provide sufficient driving-

current, with $L=4\ \mu\text{m}$, $W=30\ \mu\text{m}$. As shown in Fig. R1a, the CNT receiver can be fabricated by connecting the Pd electrode of the cascading detector to the gate-end of the n-FET. Similarly, the CNT transmitter is fabricated by connecting the source-end of the n-FET to the drain-end of the emitter as shown in Fig. R1b.

➤ **Device fabrication details:**

All devices presented in the manuscript are fabricated via a doping-free technique by selecting symmetric electrodes for CMOS devices (Ref 21) and asymmetric electrodes for optoelectronic devices (Refs 22, 23, 24, 39). Device layout patterns are defined using electron-beam lithography (EBL, Raith150-two, Germany) and all electrodes are deposited using electron-beam evaporation (EBE, Kurt J. Lesker). The metal used for fabricating the n-FET is Sc (Zhang, Z. Y. et al, 2007, Nano Lett., 7, 3603). The metals used for the cascading detector are Pd and Sc (Refs 22, 23, 24, 39). The metals for the CNT emitter are Ti and Au, in which Ti is utilized as the adhesion layer. For n-FET and cascading detector, the CNTs outside the active channels are etched using reactive ion etching (RIE, Trion Minilock-Phantom III). The insulating layer is deposited using low-temperature atomic layer deposition (Cambridge, ALD). CNTs of the emitter are assembled using the dielectrophoresis (DEP) method (Ref 30).

➤ **More scientific content**

The practical enlarged SEM image of the transmitter and receiver are given in Figs. R1a and R1b, with corresponding components being marked with white rectangle.

Revision: 1). Corresponding device structure and dimensions are added in the revised manuscript. 2). Device fabrication section has been rewritten on Page 13 and 14 of the revised manuscript, with more details given in section 1 of Supplementary Information. 3). Figures R1a and R1b are added into revised

manuscript as Figs. 1b and 2b to illustrate the scientific content in the revised manuscript.

“For all devices, the device layout patterns were defined using electron-beam lithography (EBL, Raith150-two, Germany) and all metal films were deposited using electron-beam evaporation (EBE, Kurt J. Lesker) with a standard lift-off process. For n-FET and cascading detector, CNTs outside the active channels were etched using reactive ion etching (RIE, Trion Minilock-Phantom III). The insulating layer was deposited using low-temperature atomic layer deposition (Cambridge, ALD). Detailed fabrication processes are listed in section 1 of the Supplementary Information.”

Figure R1 | **a-b**, Enlarged SEM images of CNT photoreceiver (a) and transmitter (b).

2. On line 79, “for clarity” is repeated twice, should remove one from the text.

Response: Accepted with thanks, and relevant error is corrected.

3. As for the fabrication of the vertically stacked devices, how were the patterns/layers aligned? Is there any impact from overlapping and non-overlapping regions? A comparison should be provided between the performance of the single layer devices and how they change (if at all) once they are stacked

together. From Figure 4d, it seems the performance becomes worse. Please explain.

Response: Accepted, and the text is revised.

➤ **How were the patterns/layers aligned?**

The patterns/layers are aligned using 3-points alignment marks as shown in Fig. R2a (highlighted using black squares). When we fabricate the system, all the structures are fabricated according to these three manual marks, enabling them to locate on the correct positions for alignment. Here we take the cascading Pd-Sc contact electrodes as an example to illustrate this process. In Fig. R2b, the Pd electrode pattern can be defined using the alignment marks via EBL, then using EBE to evaporate Pd film. Secondly, we repeat above processes to define the Sc contact pattern and evaporate Sc film (Fig. R2c). Similarly, if all the fabrication processes are carried out according to these same 3-points marks, then patterns/layers can be well aligned.

Figure R2 | a-c, Alignment process of the cascading detector using 3-points alignment marks.

➤ **Comparisons between the performance of the single layer devices and how they change once they are stacked together.**

As shown in Fig. R3a, the performance of the CNT transmitter is nearly the same before and after stacking. We measured the performance of individual CNT based

cascading detector as shown in Fig. R3b. After stacking, the performance of the integrated system is shown in Fig. 4d. The photovoltage of the CNT-based nine-cell cascading detector is about 82 mV under illumination power of 5.92 μW (similar to the radiated power of the CNT emitter). As a comparison, the photovoltage after stacking is 55 mV (shown in Fig. 4d). We contrast the differences of the two light sources conditions, i.e., external continuous laser and on-chip CNT emitter. In the stacked devices, the CNT emitter is located in the middle of the active detector channel with effective length of 0.5 μm . Therefore, only the middle of the detector channel is effectively illuminated by the CNT emitter as shown in Fig. 4b and inset of Fig. 4d. However, when we test the individual CNT detector, the external light sources illuminate the entire channel as shown in the inset of Fig. R3b. Experimentally, we designed such an experiment, in which we used 100-nm Ti photon isolation layer to cover the regions which were not illuminated by the emitter in the stacked devices as shown in the inset of Fig. R3c. The photovoltage of the detector became 48 mV (Fig. R3c), which is similar to the value after stacking.

Figure R3 | **a**, The emission characteristics of the CNT emitter before and after stacking. **b**, Photovoltaic response of the CNT detector. Inset is the corresponding optical image. **c**, Photovoltaic response of the CNT detector with 100-nm Ti to cover the regions where were not illuminated by the CNT emitter. Inset is the corresponding optical image.

Revision: Figures R2 and R3 are added into the Supplementary Information as Figs. S5 and S7. Following discussions are added on Page 9 of the revised manuscript.

“Corresponding alignment process is discussed in section 6 of the Supplementary Information and shown in Fig. S5.”

“When compared with well separated emitter and detector, the performance of the system remains basically the same after stacking them together (detailed discussions are given in section 7 of the Supplementary Information).”

4. More details on the FEM simulation on page 9 need to be provided, perhaps in the supplementary information.

Response: Accepted. To estimate the power from the top-layer CNT emitter on the bottom-layer detector, we simulated the radiation distribution using COMSOL Multiphysics software. The two-dimensional simulated region is surrounded by the perfectly matched layer (PML). The thickness of SiO₂, HfO₂ insulating layer and Au electrodes are 500 nm, 20 nm, 40 nm, separately. An electric dipole is utilized as the light source because CNTs are dipole-allowed emission (Refs 31, 32). Corresponding permittivity of materials in the simulations were set according to the data in the literature (Palik, E. D. Handbook of Optical Constants of Solids II; Academic Press: Orlando, 1985).

Revision: Above discussions are added into the Supplementary Information. We also moved the inset of Fig. 4d as Fig. S9, and added on Page 10 the following sentence to the revised manuscript.

“More details are discussed in section 8 of the Supplementary Information.”

5. On page 10, the authors stated that “the system response is limited by parasitic capacitances and mismatch with external measurement systems....”, please explain the details on this

Response: The response of our system is limited at ~kHz level (Fig. 1i), which can be attributed to parasitic capacitances as analyzed in detailed earlier by Zhang, P. P. et al. (2015, Nano Res., 8(3), 1005). It was demonstrated that the working frequency of the system will be lagged at ~kHz level when the system capacitance dominates. The lumped capacity C_{sys} consists of parasitic capacitances from probes, coaxial cables and measurement instruments. The delay of the system was measured to be about 40 ns by shortening the two probes directly and the system resistance R_{sys} is $\sim 5 \Omega$ according to the measured results. Therefore, the C_{sys} can be estimated as $8 \times 10^{-10} \text{ F}$ ($\tau = R_{\text{sys}} \times C_{\text{sys}}$). When involving this large capacitance, the system response speed is restricted at ~kHz level (as shown in Table 3 of Ref Zhang, P. P. et al, 2015, Nano Res., 8(3), 1005). More detailed simulation results have also been shown in the reference, which agrees with corresponding experimental ~kHz response speed.

Revision: Above discussions have been added into the Supplementary Information. The following sentence is added on Page 10 of the revised manuscript.

“Further analyses are discussed in section 5 of the Supplementary Information.”

6. It is described that “the emitted photons from the top layer emitter do not heat the conduction channels ... not affected.” Does this remain true if running for a longer time? Any further data to support this conclusion?

Response: Yes. Here we aim to emphasize that the photovoltage of the bottom-layer detector is resulted from photovoltaic effect instead of bolometric or photothermoelectric effect. Therefore, we utilized similar power density to illuminate the n-FET as shown in Fig. S11a, showing no photovoltage effect in the n-FET. As a comparison, when the same light power was illuminated on the

cascading detector, typical photovoltage and photocurrent are generated, indicating the photovoltaic nature of the integrated system. In addition, we also want to show that the emitted photons from the CNT emitter will not affect the operation of the FET-based CNT logic devices. To do this, we utilized a p-FET under the illumination power of $1.19 \mu\text{W}$ and $5.92 \mu\text{W}$ (these powers are similar to that emitted by the CNT transmitter) for ten minutes separately, and then compare its transfer characteristics with the original one. Figure R4 compares these results, showing again no obvious changes of the p-FET transfer characteristics of the FET. Thus, we thus shown that both n-FET (Fig. S11a) and p-FET (Figure R4) will not be affected by the emitted photons from the emitter, and conclude that the performance of the CNTs-based logic devices will not be affected by the CNT emitter.

Figure R4 | Long-time illumination stability of the p-FET.

Revision: Above discussions are added into the Supplementary Information. Figure R4 is added as Fig. S11b. The following illustration is added on Page 11 of the revised manuscript.

“Further analyses are discussed in section 9 of the Supplementary Information (Fig. S11b).”

7. In the double-layer 3D OEIC system shown in Fig.5, how to avoid the interference between emitters during the mapping process? Should the insulating layer be transparent for this design? In addition, it would be interesting to know if this kind of design works for more than two layers?

Response:

- **How to avoid the interference between emitters during the mapping process?**

The interference between emitters during the mapping process can be avoided by using photon isolation layer. As shown in Fig. R5, the CNT emitter can be covered by 100-nm Ti as the photon isolation layer. The calculated photon transmission probability of the 100-nm Ti is on the order of 3×10^{-3} (inset of Fig. R5b), giving rise to a low transmission. Thus, as compared with Fig. R5a, no emitted IR light can be detected as shown in Fig. R5b.

Figure R5 | **a**, EL emission characteristics of the CNT emitter. Inset is the optical image. **b**, Corresponding emission behavior with 100-nm Ti photon isolation layer covered on top of it. Inset: Left: Calculated photon transmission probability of 100-nm Ti. Right: Corresponding device optical image.

- **Should the insulating layer be transparent for this design?**

We only need to use insulating material without absorption at the communication band of the system, and transparency is only required for the intended wavelengths.

➤ **In addition, it would be interesting to know if this kind of design works for more than two layers?**

In principle, this kind of design can be effective for more than two layers. As demonstrated in previous part (Response of Review#1 Question 4), there is also upward emitted light of the CNT emitter, which has been detected by the external detector (Fig. 4c). Therefore, if we fabricated another layer of CNT detector on top of the CNT emitter, it is expected that the upward emitted energy could also be detected by another top-layer CNT detector.

Revision: Figure R5 and corresponding discussions are added into Supplementary Information as Fig. S12. The following discussions are added on Page 12 and 13 of the revised manuscript.

“In principle, the upward emission can be suppressed using photon isolation layer such as 100-nm Ti to avoid interference between emitters during the mapping process (Further analyses are discussed in section 10 of the Supplementary Information, Fig. S12).”

“thus potentially enabling the realization of high-speed and large-scale monolithic multilayer OEICs.”

8. Equivalent circuits for the AND in Fig.3 should be added into the figures.

Response: Accepted, and corresponding equivalent circuits for the “AND” gates are given in Fig. R6.

Revision: Figure R6 replaces new Figs. 3a and 3d.

Figure R6 | The “AND” gate realized with the receiver (a) and transmitter (b). Insets are the corresponding “truth table” and equivalent circuits.

Reviewer# 2

Y. Liu et al. reports a 3D optoelectronic integrated circuit fabricated with carbon nanotube dopant free electronic and optoelectronic devices. They demonstrate CNT PV receiver, a CNT IR transmitter, heterogeneous integration with logic CNT elements, and the monolithic 3D system integration.

I believe that the overall effort merits publication in Nature Communications. I am concerned about the claim in the communication speed of $10n^2$ Gb/s. I suggest this to be removed from the abstract. It is clear that the system parasitics dominate the performance of the device at kHz frequencies that are tested in this work. For 3D integrated OEIC, the engineering challenges reside in interconnect bandwidth and power. These two challenges were not addressed in this work and as such, while the fabrication and integration is exotic, I am afraid that the work would be another demonstration that ‘we can do it also with CNTs’ without a strong and complete benchmarking of performance metrics against existing technologies.

I recommend publication with minor revision.

Response: We thank the referee for the very positive comments, and suggestions are

accepted. We note that this work is only the first step to construct all CNTs based 3D integrated optoelectronic system. Our next target is to improve the system and to benchmark against existing technologies with established procedures and performance metrics. Currently, the response speed of the system is dominated by system parasitic capacitances etc. at ~kHz level as discussed in previous part (Response of Reviewer#1 Question 5). In principle, the emission speed of CNT-based emitter is ~140 ps (Ref 36) and the speed of CNT-based detector is also on the order of picosecond (Prechtel, L. et al., Nano Lett., 2010, 11.1, 269). The fast individual emitter and detector provide us with a large space to optimize the system design structure for achieving high interconnect bandwidth with low power consumption. And this is exactly what we intend to do, and we understand that it may take us a long time to achieve it.

Revision: The sentence about “communication speed of $10n^2$ Gbps” has been removed from the abstract, and replaced with the following text:

“which could enable a higher communication speed”

Reviewer# 3

The authors have prepared an electrically driven carbon nanotube (CNT)-based on-chip 3D optoelectronic integrated circuit (OEIC), which show diversified functionality such as the heterogeneous “AND” gates. This work seems to be interesting and the structure is well organized, but CNT foam structure has well-studied before, and there is not enough novelty and the results does not show significant findings.

Response: We disagree with the referee on this. While “CNT foam structure has well studied before” , novel OEICs have never been realized, and indeed the very fact that emission from electrically driven CNT emitters can be detected by CNT photovoltaic devices is an important finding, which provides the very

foundation of the OEIC and has not been demonstrated before.

There are also some questions needed to be further clarified.

1. What is the advantage of single material (CNTs)-based monolithic optoelectronic compared with CNTs-Silicon based monolithic optoelectronic?

Response: We note that we do not indeed have a “CNTs-Silicon based monolithic optoelectronic” to compare with. The unique advantage with CNT based technology is that both CMOS and optoelectronic devices can be fabricated using CNT with the same doping-free procedure, which is very simple, involving basically only deposition of symmetric (for CMOS) and asymmetrical (for optoelectronic devices) electrodes. All relevant electronic and optoelectronic components (FET, emitter, detector) can be fabricated with the same process. On the other hand, no “CNT-Silicon” technology has been demonstrated that is able to build OEICs.

2. Please list the photocurrent responsivity, electrical on/off ratio, and optical on/off of the device in this work. In order to clearly reveal the excellent performance of the as-prepared device, please compare those parameters with the past reports.

Response: We have listed photocurrent/photovoltage responsivity, electrical on/off ratio and optical on/off ratio of our devices in Table R1, and compared with that reported in Ref 29 (Kim, Y. L. et al., 2014, Nat. Photonics 8, 239). In this manuscript, the performance of our photovoltaic receiver is indeed excellent, with a very high photovoltage responsivity of 10^8 V/W, enabling utilization of photovoltage as the detection signal. Most important, this photovoltage can be readily multiplied with virtual contacts, making the photovoltage responsivity three orders of magnitude higher than that of Ref 29. While photocurrent responsivity of our device is 0.67 A/W, which is smaller than that of Ref 29 (1 A/W), it could be further improved by utilizing higher quality semiconducting

CNT parallel array to increase the current and transconductance of the FET (Brady, G. et al, Sci. Adv., 2016, 2:e1601240, 1). While the electrical on/off ratio of our device is 10^5 , which is similar to that of Ref 29, the optical on/off ratio of our device is also 10^5 , that is 10 times larger than that of Ref 29. The superior performance of our device can be attributed to the efficient photovoltage controlling on the gate-end of the n-FET as shown in Fig. 1g.

Revision: Table R1 is added into Supplementary Information as Table S1 to replace the inset of Fig. 1h of the revised manuscript.

	Photocurrent responsivity	Photovoltage responsivity	Electrical on/off ratio	Optical on/off ratio
Ref 29	1 A/W	1×10^5 V/W	10^5	10^4
This work	0.67 A/W	1×10^8 V/W	10^5	10^5

Table R1 | Comparison of the photocurrent/photovoltage responsivity, electrical and optical on/off ratio of the device with the past reports.

3. The structural and scalability are very important for the practical application of the device. How is the method presented in this work?

Response: Wafer-scale fabrication of both electronic and optoelectronic devices have been realized via the doping-free fabrication technique we used in this work, with demonstrated “structural and scalability” on 2-inch silicon wafers in Ref 23 (Chen, B. Y. et al., Nano Lett., 2016, 16.8, 5120; Liu, Y. et al, 2016, Adv. Opt. Mater., 4, 238). While there will surely be some practical problems toward industrial production, we believe that we have demonstrated the scalability of this technique on a scientific and technical level.

4. The authors just show a 2x2 array of transmitter/receiver pairs, but what is the 4x4 array of transmitter/receiver pairs? If possible, please provide the corresponding

performances.

Response: As shown in Fig. R7a, the 4×4 array of transmitter/receiver array was fabricated with uniform output photovoltage shown in Fig. R7b. Actually, as compared with the 2×2 array, the 4×4 array is only a scaled-up process. In principle, it can be scaled up to any desired n×n array.

Figure R7 | a, Optical image of the 4×4 vertical parallel transmission array. b, Corresponding photovoltage outputs.

5. The infrared wavelength of the laser is just from 1165 to 2100 nm, so what is the sensitivity/performance of this device in variable light conditions (the laser with wide wavelength)?

Response: CNT-based detector has a broadband response from ultraviolet to infrared (IR), which can indeed be turned by selecting CNTs with appropriate diameter and chirality (Itkis, Mikhail E. et al., Science, 2006, 312.5772, 413). In Ref 23, the performance of CNT detector in variable light conditions have already been studied. In this study, we are interested in the NIR band due to its potential in optical communication, and selected to use (8, 3) & (8, 4) enriched CNTs with emission central wavelengths at ~1347 nm and ~1192 nm, respectively.

Revision: We added the following illustration on Page 5 of the revised manuscript.

“and the detector responses in the whole NIR band, i.e., from 1165 nm to 2100 nm²³.”

6. The authors claimed that the emission wavelength can be readily controlled via the selection of appropriate CNTs with suitable chirality. Maybe it means that it will have different performance/conclusion if different CNTs have been used.

Response: The message we tried to send is exactly what we stated in the manuscript “the emission wavelength can be tuned by using different chirality CNTs”, which is not as suggested by the referee as “it will have different performance/conclusion if different CNTs have been used”. Different chirality CNTs have different emission wavelengths, e.g. the wavelength of “trion” induced emission of (8, 3) and (8, 4) CNTs are at 1347 nm and 1192 nm, respectively, see Fig. S6b. In principle, the response of the detector will be similar if the same light intensity is illuminated on the device. Here, we designed an experiment to prove it. We tuned the emission wavelengths of the continuous laser at 1192 nm, 1310 nm, 1347 nm and 1550 nm (all between the E_{11} and E_{22} of the high-semiconducting-purity CNTs), and ensure the illuminated power on the device is 1

mW. As shown in Fig. R8, the photocurrent and photovoltage of the detector at these four wavelengths are nearly the same with average values of 2.59 nA and 1.125 V, respectively. The standard deviation of photocurrent and photovoltage (Figure R8b) are 2.13×10^{-10} A and 0.0058 V, indicating highly uniform photo-response.

Revision: The following sentence is added on Page 6 in the revised manuscript to make sure that that we will not be understood:

“In general, the emission wavelength can be readily controlled via the selection of appropriate CNTs with suitable chirality, which can tune the communication wavelength of the transmitter.”

Figure R8 | **a**, Output characteristics of the cascading detector under the same power intensity of four different wavelengths. **b**, Corresponding photocurrent and photovoltage values of the detector.

Reviewers' comments:

Reviewer #4 (Remarks to the Author):

I believe that the authors have carefully addressed the referee's concerns. Upon addressing the following points, I recommend the manuscript for publication.

- The previous important reports related on this paper are not referred in the introduction of the main text. For example, Refs 34 and 29 are previous studies on vertically integrated optical communications and digital optoelectronics with CNTs, respectively. Please refer these reports in the introduction section of the manuscript and show the originality of the author's study.
- In Fig. 4, the top and bottom devices can be capacitively coupled because the interlayer insulator of 20-nm HfO₂ is very thin and high-k dielectrics. Therefore, the electrodes of the emitters on the top layer can work as local gate electrodes for CNTs of bottom layer under voltage for emission, and can modulate the band structure of CNTs in bottom-layer detector. Can the effect of the capacitive coupling be ruled out?
- The authors discuss about channel division multiplexing (CDM) in Fig. 5 and p. 11-13 in the main text. However, the result in Fig. 5 is not the demonstration of CDM (i.e. digital optical communication) but the demonstration of light detection between 2D arrayed transmitters/receivers under DC condition. Therefore, I suggest
 - (i) the description of CDM to be removed from the abstract.
 - (ii) Figure 5a and 5b to be removed from the manuscript.
 - (iii) the description of CDM on p. 11-13 to be removed or reduced..
- Please show the measurement conditions of Fig. 1i (light power, wavelength, light source) in the figure or figure caption.
- Typo: p. 16, "24. Yang, L.J. et al. Extremely efficient photovoltage...." => "24. Yang, L.J. et al. Efficient photovoltage...."

Reviewer #5 (Remarks to the Author):

In this manuscript, the authors demonstrated carbon nanotube based 3D monolithic optoelectronic integrated circuits (OEIC). Specifically, they presented on-chip transmitter, receiver, proof-of-concept OEICs such as heterogeneous logic gates. The results are clearly demonstrated, strongly supported and the manuscript is well written, especially after 1st round revision. The authors have answered the questions/comments raised by reviewers (#1^L) clearly (except for 1 point, see below) in detail. I believe it can be published on Nature Communications. I, personally, have some comments to be addressed before publication.

1. Are CNTs to some degree aligned produced through DEP process? If so, for any optoelectronic devices, have authors tried to demonstrate any polarization dependence?
2. In the CNT transmitter, what is the current modulation speed for CNT emitters? Probably, the authors should estimate it from Figure 1h.
3. In 3D integrated circuit, the authors claimed it as an advantage that emission peaks from the top-layer lie between E11 and E22 of CNTs from the bottom layer. Why is it? If the emission peak overlaps with the absorbance peak E11 or E22, will it help increase the generated voltage (probably the responsivity is the same) ?

4. Regarding Reviewer 1 Comment 7, I believe the reviewer was referring to the cross-talk between adjacent emitters, since the emission is not perfectly collimated. However, I don't understand how top Ti coverage can help reduce the this cross-talk. I think the authors should clarify/elaborate more.

Point-to-point response to reviewer comments:

Reviewer# 4

I believe that the authors have carefully addressed the referee's concerns. Upon addressing the following points, I recommend the manuscript for publication.

1. The previous important reports related on this paper are not referred in the introduction of the main text. For example, Refs 34 and 29 are previous studies on vertically integrated optical communications and digital optoelectronics with CNTs, respectively. Please refer there reports in the introduction section of the manuscript and show the originality of the author's study.

Response: Accepted, and short introduction to old Refs 34 and 29 (now 25,26) are given in the introduction section in the revised manuscript.

2. In Fig. 4, the top and bottom devices can be capacitively coupled because the interlayer insulator of 20-nm HfO₂ is very thin and high-k dielectrics. Therefore, the electrodes of the emitters on the top layer can work as local gate electrodes for CNTs of bottom layer under voltage for emission, and can modulate the band structure of CNTs in bottom-layer detector. Can the effect of the capacitive coupling be ruled out?

Response: Yes, the reviewer is right in pointing out that the top and bottom devices are capacitively coupled. We cannot rule out this capacitive coupling effect, but can rule out false signal from this coupling, i.e. only emission from the top layer can generate photovoltage in the bottom layer. While the capacitive coupling can result in certain changes in the bottom layer, e.g. band structure of CNTs, but not signal for the detector, i.e. photovoltage. The attached Fig. R1a and R1b show that applying a gate voltage on the top layer (without DEP CNTs) will not result in the generation of any photocurrent or photovoltage in the bottom-layer detector.

Figure R1 | **a**, Schematic of the device structure. **b**, Output characteristics.

3. The authors discuss about channel division multiplexing (CDM) in Fig. 5 and p. 11-13 in the main text. However, the result in Fig. 5 is not the demonstration of CDM (i.e. digital optical communication) but the demonstration of light detection between 2D arrayed transmitters/receivers under DC condition. Therefore, I suggest

- (i) the description of CDM to be removed from the abstract.
- (ii) Figure 5a and 5b to be removed from the manuscript.
- (iii) the description of CDM on p. 11-13 to be removed or reduced.

Revision: Accepted. (i) The description of CDM has been removed from the abstract. (ii) Figures 5a and 5b have been moved into the Supplementary Information as Figs. S13a and S13b; and Figures 5c and 5d are added as Figs. 4g and 4h. (iii) The description of CDM on p. 11-13 is largely reorganized and rewritten.

4. Please show the measurement conditions of Fig. 1i (light power, wavelength, light source) in the figure or figure caption.

Response: Accepted, and relevant parameters are added to the caption of Fig. 1i.

5. Typo: p. 16, “24. Yang, L.J. et al. Extremely efficient photovoltage....” => “24. Yang, L.J. et al. Efficient photovoltage....”

Response: Accepted with thanks.

Reviewer# 5

In this manuscript, the authors demonstrated carbon nanotube based 3D monolithic optoelectronic integrated circuits (OEIC). Specifically, they presented on-chip transmitter, receiver, proof-of-concept OEICs such as heterogeneous logic gates. The results are clearly demonstrated, strongly supported and the manuscript is well written, especially after 1st round revision. The authors have answered the questions/comments raised by reviewers (#1) clearly (except for 1 point, see below) in detail. I believe it can be published on Nature Communications.

Response: We thank the referee for the very positive comments and recommendation.

I, personally, have some comments to be addressed before publication.

1. Are CNTs to some degree aligned produced through DEP process? If so, for any optoelectronic devices, have authors tried to demonstrate any polarization dependence?

Response: The CNTs produced by DEP process are aligned to some degree as shown in Fig. S3a. But this alignment is dominating only near to the electrodes, and in the middle region between the electrodes, the CNTs in the channel are still largely randomly oriented. We carried out polarized Raman characterization for the DEP-processed CNTs as shown in Fig. R2a. These results suggest no obvious polarization effect. The CNTs in the bottom layer for the detector are in the form of random network, and measurements show no dependence on polarization, as shown in Fig. R2b.

Figure R2 | Polarized Raman characteristics of (a) DEP-processed CNTs of the top layer, and (b) random network of CNTs of the bottom layer.

- In the CNT transmitter, what is the current modulation speed for CNT emitters?
Probably, the authors should estimate it from Figure 1h.

Response: In the CNT transmitter, the current modulation was determined by the n-FET, whose response speed has been analyzed by Zhang, P. P. et al. early in 2015 (Nano Res., 8, 1005). It was demonstrated that the working frequency of the n-FET will be lagged at ~kHz level when the system capacitance dominating. In Fig. 2h, the measurement was made very slowly, and the figure is thus not suitable for estimating the response. A more appropriate measurement is shown in Fig. R3, suggesting clearly that the results agree well with our early analyses (Zhang, P. P. et al., 2015, Nano Res., 8(3), 1005).

Figure R3 | Modulated emission speed of the CNT transmitter.

- In 3D integrated circuit, the authors claimed it as an advantage that emission

peaks from the top-layer lie between E_{11} and E_{22} of CNTs from the bottom layer. Why is it? If the emission peak overlaps with the absorbance peak E_{11} or E_{22} , will it help increase the generated voltage (probably the responsivity is the same) ?

Response: In this experiment, we design to have the emission of the top-layer emitter falls between the E_{11} and E_{22} absorption peaks of the bottom-layer detector (Fig. S6b), making sure that the emission from the top layer would be detected by the bottom-layer detector. If the emission peak overlaps with the absorbance peak E_{11} or E_{22} of the bottom detector channel, resonant response will occur which may lead to larger photocurrent. However, the fabrication of our system involves over 20 steps, which often results in certain degree shift of the emission peaks (due to the varying dielectric environment), and we thus focused on achieving response, but not the best response.

4. Regarding Reviewer 1 Comment 7, I believe the reviewer was referring to the cross-talk between adjacent emitters, since the emission is not perfectly collimated. However, I don't understand how top Ti coverage can help reduce the this cross-talk. I think the authors should clarify/elaborate more.

Response: I agree that the not perfectly collimated emission might in principle result in certain cross-talk effect, but in our case since the pairs of emitter/detector are very closed packed ($\sim 30\text{nm}$) and well separated from each other (over μm), it is thus unlikely to result in cross-talk between adjacent channels (pairs of emitter/detector). The emitted but not completely absorbed light, e.g. the upward emitted light might potentially affect other parts of the OEIC, and should be isolated. Ti coverage is shown to be able to prevent the light from escaping into other parts of the devices, suggesting that the electrodes around the active parts of the emitters/detectors may help to reduce the cross-talk between adjacent channels (pairs of emitter/detector).

100-nm Ti is utilized as proof-of-concept photon isolation layer by virtue of its

low transmittance (left inset of Fig. S12b) to avoid the interference resulted by the upward emission between adjacent emitters during the mapping process. In this sense, the transmission of 100-nm Au layer is better than Ti because of its transmission can be below 10^{-4} as shown in Fig. R4b. Therefore, we can utilize Au as the reflector of the vertical Fabry-Perot cavity to realize effectively collimating of the emission in principle.

Figure R4 | Transmission of 100-nm Ti film (a) and 100-nm Au film (b).

REVIEWERS' COMMENTS:

Reviewer #5 (Remarks to the Author):

I think the authors have clearly answered my and other reviewer's comments/questions. I think it is suitable for publication now.